


Atmospheric
Measurement
Techniques

# Airborne measurements of particulate organic matter by proton-transfer-reaction mass spectrometry (PTR-MS): a pilot study

**Felix Piel**[1,2], **Markus Müller**[1], **Tomas Mikoviny**[3], **Sally E. Pusede**[4], **and Armin Wisthaler**[2,3]

[1]Ionicon Analytik GmbH, 6020 Innsbruck, Austria
[2]Institute for Ion Physics and Applied Physics, University of Innsbruck, 6020 Innsbruck, Austria
[3]Department of Chemistry, University of Oslo, 0315 Oslo, Norway
[4]Department of Environmental Sciences, University of Virginia, Charlottesville, VA 22904-4123, USA

**Correspondence:** Armin Wisthaler (armin.wisthaler@kjemi.uio.no)

**Abstract.** Herein we report on the first successful airborne deployment of the CHemical Analysis of AeRosol ONline (CHARON) particle inlet which allowed us to measure the chemical composition of atmospheric submicrometer particles in real time using a state-of-the-art proton-transfer-reaction time-of-flight mass spectrometry (PTR-ToF-MS) analyzer. The data were collected aboard the NASA DC-8 Airborne Science Laboratory on 26 June 2018 over California in the frame of NASA's Student Airborne Research Program (SARP). We show exemplary data collected when the airplane (i) shortly encountered a fresh ( < 1 h old) smoke plume that had emanated from the Lions Fire in the Sierra Nevada, (ii) intercepted a particle plume emitted from an amine gas treating unit of a petroleum refinery close to Bakersfield, (iii) carried out a spatial survey in the boundary layer over the San Joaquin Valley and (iv) performed a vertical profile measurement over the greater Bakersfield area. The most important finding from this pilot study is that the CHARON PTR-ToF-MS system measures fast enough to be deployed on a jet research aircraft. The data collected during 3 to 15 s long plume encounters demonstrate the feasibility of airborne point or small area emission measurements. Further improvements are, however, warranted to eliminate or reduce the observed signal tailing (1/e decay time between 6 and 20 s). The fast time response of the analyzer allowed us to generate highly spatially resolved maps (1–2 km in the horizontal, 100 m in the vertical) of atmospheric particle chemical constituents. The chemical information that was extracted from the recorded particle mass spectra includes (i) mass concentrations of ammonium, nitrate and total organics; (ii) mass concentrations of different classes of organic compounds (CH vs. CHO vs. CHN vs. CHNO compounds; monoaromatic vs. polyaromatic compounds); (iii) aerosol bulk average $\overline{O:C}$ and $\overline{H:C}$[TS1] ratios; (iv) mass concentrations of selected marker molecules (e.g., levoglucosan in particles emitted from a wildfire, an alkanolamine in particles emitted from a petroleum refinery) and (v) wildfire emission ratios ($\Delta$total organics/$\Delta$CO $= 0.054$; $\Delta$levoglucosan/$\Delta$CO $= 7.9 \times 10^{-3}$; $\Delta$vanillic acid/$\Delta$CO $= 4.4 \times 10^{-4}$ and $\Delta$retene/$\Delta$CO $= 1.9 \times 10^{-4}$; all calculated as peak area ratios, in grams per gram). The capability of the CHARON PTR-ToF-MS instrument to chemically characterize submicrometer atmospheric particles in a quantitative manner, at the near-molecular level, and in real time brings a new and unprecedented measurement capability to the airborne atmospheric science community.

## 1 Introduction

Proton-transfer-reaction mass spectrometry (PTR-MS) is a well-established technique for online and real-time detection of organic trace gases in the Earth's atmosphere (de Gouw and Warneke, 2007; Yuan et al., 2017; and references therein). One of the main advantages of PTR-MS is the rapidity at which the air can be analyzed. Measurement frequencies of 1 to 10 Hz make PTR-MS instruments ide-

ally suited for deployment on fast moving platforms such as airplanes. The first airborne deployment of a PTR-MS analyzer dates back to 1998 (Crutzen et al., 2000). Nowadays, most in situ tropospheric chemistry payloads of research aircraft include a PTR-MS instrument. Our group has been flying PTR-MS analyzers on NASA airplanes (DC-8, P-3B, C-130) since 2008. We developed and deployed the first airborne proton-transfer-reaction time-of-flight mass spectrometry (PTR-ToF-MS) instrument (Müller et al., 2014). More recently, we conceived and built the CHARON (CHemical Analysis of AeRosol ONline) particle inlet which enables PTR-MS analyzers to also measure and chemically characterize submicrometer particles in real time (Eichler et al., 2015). In ground-based studies, we have successfully used the CHARON inlet in combination with a PTR-TOF 8000 analyzer for measuring (i) particle mass concentrations of total organics, ammonium and nitrate; (ii) aerosol bulk average $\overline{O:C}$ and $\overline{H:C}$ ratios; (iii) mass concentrations of selected tracers (e.g., levoglucosan, pyrene) and (iv) mass concentrations associated with different source-related multivariate factors (e.g., biomass burning, traffic and smoking) (Müller et al., 2017). In 2018, the opportunity arose to test-fly a CHARON inlet on the NASA DC-8 in the frame of NASA's Student Airborne Research Program (SARP). In this work, we will show exemplary results from the first airborne CHARON PTR-ToF-MS deployment and discuss possibilities and current limitations.

## 2 Materials and methods

### 2.1 Test flight description

The data presented herein were collected on a flight that the NASA DC-8 carried out in California on 26 June 2018 (DC-8 flight number 1271). The flight track is shown in the Supplement (Fig. S1). We note that, while SARP provided an excellent opportunity for test-flying the CHARON inlet, the flight plan was neither designed for best characterizing the instrument's performance nor for addressing specific science questions related to the chemical composition and transformation of organic particles in the atmosphere. Herein we present data from three selected flight segments. During one flight segment, the NASA DC-8 shortly penetrated the smoke plume emanating from the Lions Fire (Wikipedia, 2018) in the Sierra Nevada (see photograph shown in Fig. S2). The fire was burning in red fir and brush, in an area of heavy blowdown from a 2011 wind event (InciWeb, 2018). The airplane overflew the fire in the plume direction and dropped into the plume at a safe distance, 14.3 km downwind of the source. This converts into a plume travel time of $\sim 45$ min if we assume an average wind speed of 5 m s$^{-1}$ (as measured on the DC-8). The plume was encountered at 13:04 local time (20:04 UTC), at 37.6058 (37°36′20.862″) N and 118.9210 (118°55′15.726″) W, at an altitude of 4630 m (2160 m a.g.l.).

The static air temperature at this altitude was 0.4 °C. Due to safety concerns (reduced visibility over mountainous terrain), the airplane only skimmed the plume edge for less than a minute.

In the San Joaquin Valley, the NASA DC-8 flew race track patterns in the boundary layer over the greater Bakersfield area which include oil and gas extraction areas to the N and NW of the city and agricultural areas to its S and SE. The first race track pattern included a missed approach (down to 50 m a.g.l.) at the Meadows Field Airport, which is located approximately 5 km NW of downtown Bakersfield.

### 2.2 Sampling inlet

Ambient air was sampled into the DC-8 through a forward facing shrouded solid diffuser inlet known in the literature as the UH/LARGE inlet. The inlet and its characteristics have been described in detail by McNaughton et al. (2007). The authors of that study found that the inlet transmits particles with unit efficiency up to particle sizes in the micrometer (μm) range. The flow rate at the inlet tip was maintained at isokinetic values by measuring the sampling mass flow and automatically throttling a butterfly valve based on aircraft true airspeed (Brock et al., 2019). Our instrument was connected to the sample plenum of the UH/LARGE inlet via a 7.6 mm inner diameter sampling port, a reducing union connector and a 1.5 m long passivated stainless steel tube of 4.57 mm inner diameter. At the end of this sampling tube, we placed a so-called pressure-controlled inlet (Bahreini et al., 2008), i.e., a small volume between a 1.1 mm inlet orifice and a 3.3 mm exit orifice towards the CHARON inlet. This volume was kept at constant pressure using a variable bypass flow into a digital pressure controller (Bronkhorst High-Tech BV, Ruurlo, the Netherlands CE1). For boundary layer sampling, the pressure was kept at 800 mbar. For sampling the biomass burning plume at 4600 m altitude, we had to lower the pressure to 610 mbar. The pressure-controlled inlet was designed to maintain isokinetic or near-isokinetic sampling conditions from the sample plenum and into the CHARON inlet for ambient pressures between 600 and 1000 mbar. A sketch of the complete inlet system is given in Fig. S3.

### 2.3 CHARON particle inlet

The CHARON particle inlet for PTR-ToF-MS instruments was described in detail elsewhere (Eichler et al., 2015) and we will only give a brief description here. The CHARON inlet consists of a gas-phase denuder, an aerodynamic lens and a particle vaporizer. Sampling air is first passed through an activated carbon monolith denuder, which removes gaseous organics and transmits particles larger than 50 nm. The aerodynamic lens efficiently enriches particles in the 150 to 1000 nm range in the instrument subsampling flow, with typical enrichment factors ranging from 35 to 45. Particles in the 60 to 150 nm range are less efficiently concentrated in

the subsampling flow. The enrichment factor is determined in an external calibration with size-selected particles that are counted using a condensation particle counter (CPC). Due to a misadjustment, the enrichment factor was only 12 (8 for the Lions Fire plume encounter due to the reduced sampling pressure) for the study presented herein. The diameter of the critical orifice at the entrance of the aerodynamic lens was increased by 10 % to account for the lower sampling pressure from the pressure-controlled inlet. Particles were vaporized at a temperature of 150 °C and an absolute pressure of 8 mbar. 150 °C is the lowest temperature at which ammonium sulfate (AS) particles fully evaporate in the CHARON system (i.e., the ammonium signal in the PTR-MS analyzer levels off when laboratory-generated AS particles are sampled and the vaporizer temperature is gradually increased). It is important to note that in laboratory studies we have hitherto not observed any thermal breakdown of analytes (Stark et al., 2017) in the vaporizer, except for one organic peroxide. For determining the instrumental background, the sampling flow into the CHARON inlet was periodically diverted through a HEPA filter (ETA filter model HC01-5N-B, Aerocolloid LLC, Minneapolis, MN, USA) upstream of the denuder. A scroll pump (Varian TriScroll 600, Agilent Inc., Santa Clara, CA, USA) was used for maintaining a ∼ 500 sccm sampling flow through the CHARON inlet.

## 2.4 PTR-ToF-MS

The airborne PTR-ToF-MS instrument used for this study was described in detail by Müller et al. (2014). Since the time of publication of this method paper, the instrument has been upgraded with an ion funnel at the end of the drift tube (ION BOOSTER, Ionicon Analytik, Innsbruck, Austria) and a hexapole interface (ION GUIDE, Ionicon Analytik, Innsbruck Austria) to the orthogonal acceleration region of the time-of-flight analyzer. These changes have resulted in a 15 to 20 fold increase in sensitivity (Schiller, 2018). In addition, geometrical changes were made for improving the mass resolving power ($m/\Delta m$; ∼ 4500 for signals with $m/z > 100$). With these hardware changes, the setup of the airborne PTR-ToF-MS instrument was identical to that of a commercial PTR-TOF 4000 X2 analyzer (Ionicon Analytik, Innsbruck, Austria). Unfortunately, a short circuit in the ion extraction optics led to a deteriorated instrument performance during the test flight (∼ 65 % loss in sensitivity and ∼ 40 % loss in mass resolving power). In an attempt to resolve the problem, we opened the system the day before the test flight, which significantly increased the instrumental background. Typical detection limits ($2\sigma$, 1 Hz) of the CHARON PTR-ToF-MS instrument without hardware problems are ∼ 1 ng sm$^{-3}$ for an individual signal. During the test flight, the analyzer exhibited a 3–5 ng sm$^{-3}$ limit of detection for most signals due to the lower enrichment factor and the lower sensitivity. The elevated background on some low $m/z$ signals increased the associated detection limit to double-digit levels CE2 (ng sm$^{-3}$). The PTR-ToF-MS instrument was operated in the $H_3O^+$ mode and the drift tube was kept at a temperature of 120 °C, a pressure of 3.0 mbar and a reduced electric field strength of 60 Td (1 Td = $10^{-17}$ V cm$^2$). The dehumidifying action of the gas-phase denuder and the relatively high drift tube temperature prevented both reagent and product ions from hydrating at the relatively low $E/N$ of 60 Td (Gkatzelis et al., 2018). Mass spectra were recorded in the $m/z$ 6 to 622 range at a frequency of 1 Hz. Trace amounts of 1,3-diiodobenzene (detected at $m/z$ 204.943 and 330.848) were continuously added to the drift tube for mass axis calibration. A dynamically diluted certified calibration gas mixture (Apel Riemer Environmental Inc., Miami, FL, USA) containing a set of 12 hydrocarbons in equimolar concentrations (methanol, acetonitrile, acetaldehyde, acetone, dimethyl sulfide, isoprene, methylethylketone, benzene, toluene, $m$-xylene, 1,3,5-trimethylbenzene and $\alpha$-pinene) was used for characterizing the instrumental response and mass discrimination in the low $m/z$ range. Calibrations with high molecular weight compounds were carried out for the determining the discrimination of high $m/z$ signals.

## 2.5 PTR-ToF data analysis

The PTR-TOF data analyzer (Müller et al., 2013) was used for the initial data analysis (mass axis calibration, iterative residual peak analysis and quantification, and assignment of molecular sum formula to $m/z$). Molecular formulas were assigned based on the measured accurate mass and, where possible, the observed isotope pattern. Further analyses were performed with custom MATLAB scripts (The MathWorks Inc., Natick, MA, USA). The quantification procedure that we currently use for CHARON PTR-ToF-MS data has been described in detail by Müller et al. (2017). We assume that protonation occurs at the collision rate between $H_3O^+$ reagent ions and the respective analyte molecules. The collision rate is calculated according to the parametrization proposed by Su (1998) which uses three properties of the analyte molecule as input parameters: (i) its molecular weight which is assumed to be the measured $m/z$ minus 1, (ii) its molecular polarizability which is calculated from the elemental composition using the parametrization proposed by Bosque and Sales (2002) and (iii) its dipole moment which is assumed to be 0.3 and 2.75 for pure and substituted hydrocarbons, respectively. The estimated accuracy of this quantification method is ±30 %. In the Müller at al. (2017) study we have also shown that CHARON PTR-ToF-MS mass spectra can be quantitatively converted into a total organic mass concentration and bulk average $\overline{O:C}$ and $\overline{H:C}$ ratios. In this early work, we did, however, not consider a small but significant bias due to the loss of neutral (and thus undetected) fragments from the protonated analyte molecules. We have, in the meantime, investigated the fragmentation of a variety of analytes (19 carboxylic and dicarboxylic acids, 4 sac-

charides and 2 polyalcohols) and developed correction algorithms for the total organic mass concentration as well as for the bulk average $\overline{O:C}$ and $\overline{H:C}$ ratios (Leglise et al., 2019). Since carboxylic and dicarboxylic acids are not the main constituents of fresh wood smoke, the data presented for the smoke plume may suffer from a small quantitative bias. Yassine et al. (2014) have recently introduced a new parameter, the aromaticity equivalent $X_c$, for determining the number of aromatic rings from a molecular formula that includes heteroatoms. For organic compounds containing a given number (#) of C, H, N and O atoms, $X_c$ is calculated as follows:

$$X_C = \frac{2\#C + \#N - \#H - 2m\#O}{RDBE - m\#O} + 1,$$

$$\text{with } RDBE = 1 + \frac{1}{2}\left(2\#C - \#H + \#N\right), \tag{1}$$

where RDBE is an abbreviation for ring and double bond equivalent and $m$ is the fraction of oxygen atoms involved in the $\pi$-bond structure of the compound. It has been shown in previous work that oxygenated monoaromatic species in fresh wood smoke are mostly methoxyphenols and substituted methoxyphenols (e.g., Bertrand et al., 2018). For simple and alkylated methoxyphenols $m$ equals 0; for substituted methoxyphenols $m$ increases to maximally 0.33 if there is only a single substituent group containing one $\pi$-bonded oxygen atom (e.g., vanillin). We have thus assumed the two extreme cases, $m = 0$ and $m = 0.33$, for classifying oxygenated species. $X_c$ was calculated for each of the elementally resolved $m/z$ signals and classified the signals as monoaromatic ($2.5 \leq X_c < 2.71$) and polyaromatic ($X_c \geq 2.71$) (for details see Yassine et al., 2014). "Monoaromatics" and "polyaromatics" are the summed mass concentrations associated with these signals, respectively. Polycyclic aromatic hydrocarbons (PAHs) are not generally considered to contain heteroatoms or carry substituents. "PAHs" are thus the summed mass concentration of all $C_xH_y^+$ signals with RDBE $\geq 7$. It is important to note that the aromaticity equivalent calculation does not classify simple furans as aromatic. Furans are abundantly emitted from fires in the gas phase (Koss et al., 2018), but their concentrations in primary emitted particles appears to be small (e.g., Oros and Simoneit, 2001; Schauer et al., 2001; Fine et al., 2002). Substituted furans are classified as aromatic for the $m = 0.33$ case, if the substituent group contains a $\pi$-bonded oxygen atom (e.g., furfural or 5-hydroxymethylfurfural).

## 2.6 Ancillary data

Carbon monoxide (CO) was measured at 10 Hz by a fast commercial $N_2O$/CO Analyzer (Los Gatos Research, San Jose, CA, USA.) with an observed $2\sigma$ precision of 0.1 ppb at 1 s averaging. The flow rate was approximately 30 slpm, which was achieved using a dry scroll pump (XDS35iC, Edwards, Burgess Hill, UK). The detection cell $e$-folding residence time was determined in the laboratory to be $0.20 \pm$ 0.05 s. The gas sampling system of the instrument was customized to maintain a constant cell pressure using a proportioning valve and a proportional–integral–derivative (PID) controller as described in Wolfe et al. (2018). Dry CO data were adjusted with a laboratory-generated calibration slope and agreed with an onboard Picarro instrument (3 s time resolution) within 3 %.

## 3 Results and discussion

### 3.1 Plume measurements

In this section, we will demonstrate that the CHARON PTR-ToF-MS instrument is fast and sensitive enough for measuring and chemically characterizing pollution plumes from a jet research aircraft.

### 3.1.1 Wildfire plume

Figure 1 includes two graphs showing the data that were obtained when the NASA DC-8 shortly penetrated the smoke plume emanating from the Lions Fire in the Sierra Nevada over California. Figure 1a shows the 1 Hz averaged time trace of carbon monoxide (CO). The DC-8 shortly penetrated the fresh smoke plume three times, resulting in three CO peaks of 15, 15 and 3 s width and maximum mixing ratios exceeding 800, 600 and 300 ppb, respectively. For safety reasons, the DC-8 only skimmed the plume edge which explains why the observed CO enhancements were below 1 ppm. Figure 1b shows the 1 Hz time traces of the 10 CHARON PTR-ToF-MS signals (in raw counts per seconds, cps) with the highest mass loadings. The 90 s signal average recorded immediately before the plume penetrations was subtracted from each of the signals to yield the excess signal in the plume. Apart from this background subtraction, no other signal processing has been applied. As explained in the Methods section, the instrument suffered from an elevated background during this flight which explains the relatively high noise in some of the signals (e.g., $m/z$ 85.029) outside the plume. These near-raw data illustrate that the three maxima seen by the CO analyzer at 1 Hz are also resolved by the CHARON PTR-ToF-MS system, although with a somewhat slower time response. For most of the $m/z$ signals shown the second peak is higher than the first, as opposed to the CO trace for which the first peak is higher. This could be a real difference in chemical composition, but is probably caused by analyte losses to instrumental surfaces during the initial phase of the plume encounter. We also note that all PTR-ToF-MS signals do not drop as pronouncedly as CO between the different peak maxima. Finally, the tailing of the PTR-ToF-MS signals after the plume penetrations is more pronounced than for the CO signal. Double exponential fits applied to the 10 $m/z$ signals after the second maximum yield 1/e response times in the range between 6.2 and 20.3 s. We have shown in previous work (see Supplement of Müller et al., 2016) that ana-

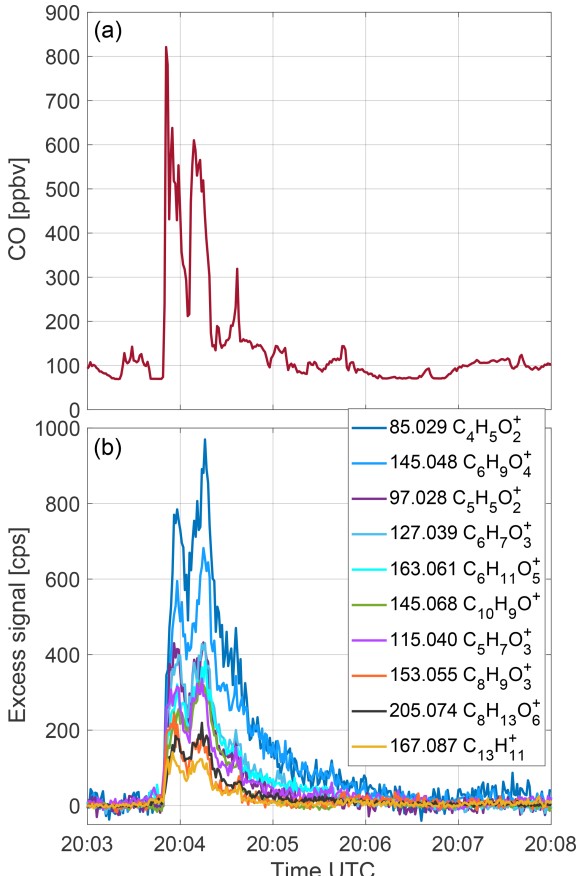

**Figure 1.** The 1 Hz time series of the **(a)** carbon monoxide (CO) mixing ratio and the **(b)** excess signal (in raw counts per seconds; 90 s pre-plume signal average subtracted) of the 10 ions ($m/z$, sum formula) with the highest mass loadings as recorded by the airborne CHARON PTR-ToF-MS instrument during a wildfire plume penetration over the Sierra Nevada in California.

lyte molecules adsorbed to instrumental surfaces are quantitatively recovered during desorption. Emissions can thus be correctly quantified if the signal tail is included in area under the peak calculations. All signals drop to pre-plume back-
5 ground levels in less than 3 min, which is the typical time the DC-8 requires to turn and resample a plume in a meander-type flight pattern. If plumes are encountered on shorter time scales (e.g., when sampling from a small aircraft or in the case of fires being closely adjacent), data quality will suffer.
The signal tailing needs to be eliminated or reduced to make the CHARON PTR-ToF-MS instrument more generally suitable for airborne applications. Test measurements by the instrument manufacturer with a yet undisclosed treatment of all wetted stainless steel surfaces indicate that the response time
can be reduced by at least a factor of 2 compared to the data shown here (Piel et al., 2019). The full mass spectrum averaged over the period including the two initial plume maxima is shown in Fig. S4.

Figure 2 shows times series of a set of processed data. The black and the red traces in Fig. 2a show the uncorrected and
20 corrected excess total organic mass concentrations, respectively. The fragmentation correction (see Methods section) increases the total organic mass concentration by $\sim 24\,\%$. The corrected bulk average $\overline{O:C}$ and $\overline{H:C}$ ratios are 0.45 (0.42) TS2 and 1.57 (1.29), respectively. The values given in
parenthesis are the uncorrected values. The observed bulk average $\overline{O:C}$ and $\overline{H:C}$ ratios resemble typical values observed for biomass burning organic aerosol ($\overline{O:C} = 0.25$–$0.55$; $\overline{H:C} = 1.55$–$1.94$; Aiken et al., 2008 TS3; Canagaratna et al., 2015; Collier et al., 2016). The organic aerosol mass
to CO ratio ($\Delta OA/\Delta CO$; calculated as peak area ratios, in grams per gram observed during the plume penetration was 0.054. This is again similar to what has been reported in the literature (Cubison et al., 2011 and references therein).

The average mass spectrum depicted in Fig. S4 shows
that most of the analyte ion signals were observed in the $m/z$ 57 to 205 range. In this range, $m/z$ signals can be elementally resolved even if measured at a somewhat reduced mass resolution (see Methods section). We were able to elementally resolve 225 $m/z$ signals in the mass spectrum ac-
counting for $\sim 75\,\%$ of the total organic mass and grouped them into four ion classes ($C_xH_y^+$, $C_xH_yO_z^+$, $C_xH_yN_z^+$ and $C_wH_xN_yO_z^+$). We note that these ion classes are commonly referred to in the mass spectrometry literature as CH, CHO, CHN and CHNO compound classes, although the elemental
composition of an ion does not necessarily reflect the elemental composition of its neutral precursor molecule (e.g., when a singly hydroxylated molecule ejects a $H_2O$ molecule upon protonation). Assuming that such fragmentation processes only cause a minor bias, the two denominations can
be used interchangeably. Figure 2a shows the 1 Hz time trace of the mass concentrations associated with the four ion classes. Because of a high instrumental background on several signals, the data were smoothed (10-point moving average) outside the plume. The relative contributions from
$C_xH_yO_z^+$, $C_xH_y^+$, $C_wH_xN_yO_z^+$ and $C_xH_yN_z^+$ ions to the total organic mass were 61 %, 7 %, 7 % and 1 %, respectively. The predominance of oxygenated hydrocarbon ions does not come as a surprise. Wood mainly consists of CHO polymers (cellulose, hemicellulose and lignin), whose depoly-
merization during combustion is known to form smaller oxygenated hydrocarbon species (e.g., Collard and Blin, 2014). The $C_xH_y^+$ ion class includes small hydrocarbon ion fragments that cannot be linked to specific neutral precursors or classes of precursors. It also includes aromatic and polyaro-
matic ions, whose neutral precursors are known to be formed from wood combustion (e.g., Rogge et al., 1998). Within the $C_wH_xN_yO_z^+$ class, we did not detect any nitrocatechols and methyl-nitrocatechols which are typical biomass burning secondary organic aerosol tracers (Iinuma et al., 2010;
Finewax et al., 2018). These species would be detected in their protonated form by PTR-ToF-MS (unpublished data from our laboratory). We do not anticipate photochemically

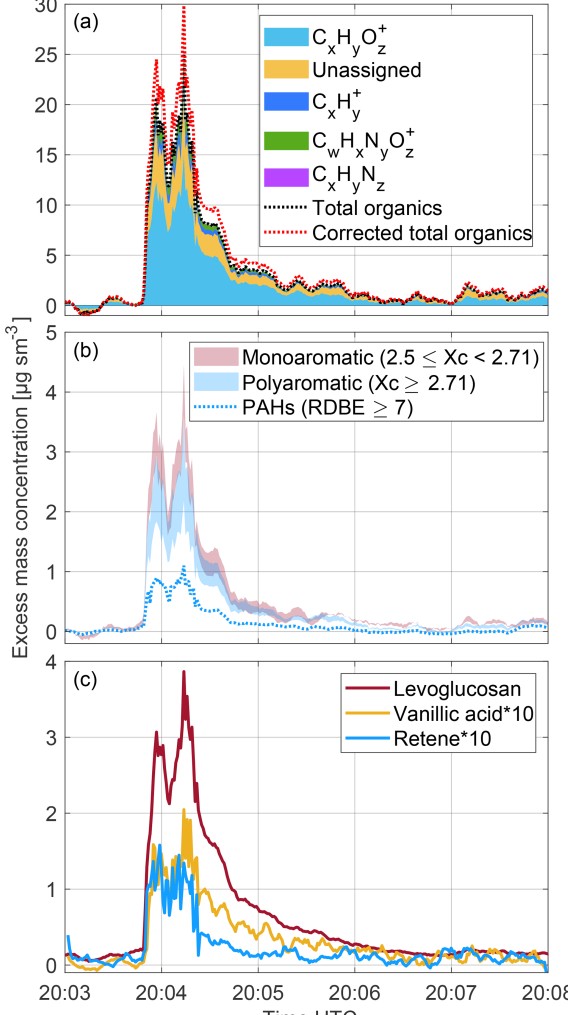

**Figure 2.** The 1 Hz time series of excess mass concentrations of **(a)** total uncorrected organics and total fragmentation-corrected organics as well as CH, CHO, CHN and CHNO compound classes; **(b)** monoaromatic and polyaromatic compounds together with PAHs (Sect. 2.5 should be consulted for details and limitations) and **(c)** levoglucosan, vanillic acid and retene as measured by the CHARON PTR-ToF-MS instrument during a wildfire plume penetration over the Sierra Nevada in California. A 10 s moving-average filter has been applied to data outside the plume.

formed nitro or nitroxy compounds being present to any significant extent since the plume was sampled in the immediate vicinity of the fire. The small contribution from $C_xH_yN_z^+$ ions is also expected since the nitrogen content of wood is small (typically $< 1\%$; Vassilev et al., 2010 and references therein).

Figure 2b shows the 1 Hz time trace of the total mass concentrations associated with monoaromatic and polyaromatic ions, respectively, as well as ions generated from polycyclic aromatic hydrocarbons (PAHs). The reader is cautioned to observe the definitions, assumptions and limitations

of our classification method given in Sect. 2.5. Monoaromatics (mostly composed of methoxyphenols and substituted methoxyphenols) account for 14 %–18 % of the total organic mass concentration, 8 %–13 % of the total mass is associated with polyaromatics (hydroxy- and dihydroxynaphthalene being the most abundant species), and PAHs account for 4.7 % of the total mass. The carbon atom and aromaticity equivalent distributions are shown in Fig. S5.

The association the $m/z$ signals in the CHARON PTR-ToF-MS mass spectrum to specific chemical compounds is a challenge. We have hitherto neither performed any laboratory biomass burning experiments nor compared CHARON PTR-ToF-MS data with data from more specific offline analytical methods. Based on a careful literature study on wood fire emissions (e.g., Oros and Simoneit, 2001; Schauer et al., 2001; Fine et al., 2002; Jen et al., 2019), laboratory calibration experiments with pure compounds and an assessment of potential mass spectral interferences, at this stage we only feel confident to report quantitative time series of three compounds, levoglucosan, vanillic acid and retene (Fig. 2c). Levoglucosan is known to be the most abundant organic compound in primary wood-fire emitted particles (Simoneit, 2002) and it exhibits a characteristic quadruplet peak pattern ($m/z$ 85.029, 127.039, 145.048 and 163.062) in the PTR-ToF-MS mass spectrum. Also, it has been shown that its isomers mannosan and galactosan are emitted in much lower yields from wood fires (Oros and Simoneit, 2001; Schauer et al., 2001; Fine et al., 2002). Figure 2c shows the 1 Hz time trace of levoglucosan as recorded by the CHARON PTR-ToF-MS instrument during the Lions Fire plume penetration. The observed $\Delta$levoglucosan/$\Delta$CO ratio (calculated as peak area ratios, in grams per gram was $7.9 \times 10^{-3}$. Vanillic acid and retene are also known to be emitted from wood fires; the latter in particular from coniferous wood combustion. The observed $\Delta$vanillicacid/$\Delta$CO and $\Delta$retene/$\Delta$CO ratios (again calculated as peak area ratios, in grams per gram were $4.4 \times 10^{-4}$ and $1.9 \times 10^{-4}$, respectively. Vanillic acid and retene exhibit little or no fragmentation in the PTR-ToF-MS instrument and the literature suggests that interferences from isomers are small. Table S1 lists $\sim 50$ $m/z$ signals that we tentatively assigned to chemical species previously detected in wood smoke. More laboratory validation work is needed for assessing which signals are specific for certain chemical compounds.

### 3.1.2 Industrial plume

Figure 3 shows the CHARON PTR-ToF-MS data that were obtained when the NASA DC-8 passed 0.85 km downwind of a petroleum refinery located in Kern County, SE of Bakersfield. The insert shows that the predominant wind direction was SSW. The airplane was flying at an altitude of 630 m a.g.l. Three atypical organo-nitrogen ions, $C_4H_{10}NO^+$ ($m/z$ 88.076), $C_6H_{14}N^+$ ($m/z$ 100.112) and $C_6H_{16}NO^+$ ($m/z$ 118.122) appeared in the mass spectra for $\sim 12$ s

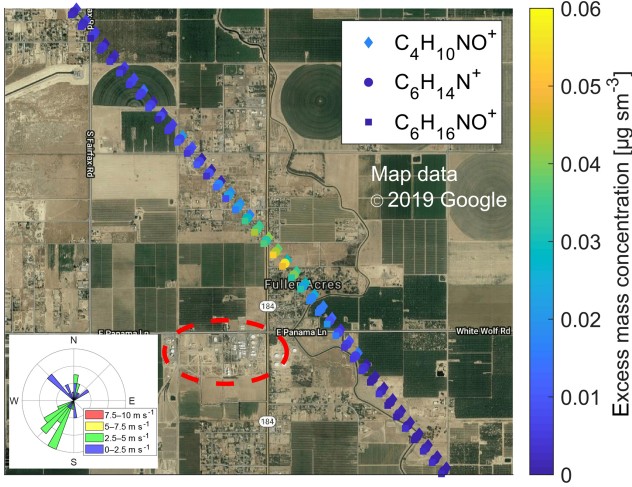

**Figure 3.** Spatial plot showing the concentration enhancements of $C_4H_{10}NO^+$ ($m/z$ 88.076), $C_6H_{14}N^+$ ($m/z$ 100.112) and $C_6H_{16}NO^+$ ($m/z$ 118.122) ions (data points slightly displaced for better visibility) in the particle mass spectra recorded at 1 Hz in a plume emanating from a petroleum refinery (circled in red) located in Kern County, SE of Bakersfield, California. The insert shows that the predominant wind direction during this flight transect was from the SSW. In the main text, we link these observations to the emission of an alkanolamine used for gas purification.

(equivalent to flight distance of 1.6 km) in the immediate downwind region of the refinery. Amines are widely used in petroleum refineries (including the one highlighted in red in Fig. 3) for gas purification (gas sweetening) and as corrosion inhibitors. Our research group has long-term experience in measuring atmospheric emissions from industrial amine scrubbing units (mostly confidential work, selected data are reported in Morken et al., 2017 and Zhu et al., 2018). $N$, $N$-diethylethanolamine typically generates the three $m/z$ signals listed above, which correspond to the $[M + H - C_2H_6]^+$, $[M + H - H_2O]^+$ and $[M + H]^+$ ions, respectively (unpublished data from our laboratory). The relative abundances of the two fragment ions of protonated $N$, $N$-diethylethanolamine are, however, much lower than observed in the plume. We may thus have observed an isomeric alkanolamine, or additional neutral decomposition products (e.g., morpholine) that were co-emitted. Amines are often released from scrubbing units in the particulate form (mist) as aminium sulfates (or as aminium nitrates). Volatile amines may also be released as gases which rapidly form particles in the atmosphere via acid–base reactions with nitric and sulfuric acid (Tan et al., 2018). To the best of our knowledge, this is the first time that aminium particles have been detected in the atmosphere downwind of an industrial facility and not directly at the stack. We are also not aware of any previous organically speciated particle measurements in which a small industrial point source was characterized from a fast jet research aircraft.

## 3.2 Spatial pollution mapping

Figure 4 shows the geographic distributions of ammonium, nitrates and total organic mass concentrations as measured during the late morning (Fig. 4a, c, e) and early afternoon (Fig. 4b, d, f) hours, respectively, in the boundary layer of the San Joaquin Valley. The inserts in Fig. 4a and b show that the prevalent wind direction shifted from SW to NW between the morning and afternoon measurements. Individual points represent 10 s data averages. In the morning, elevated levels of ammonium and nitrate were observed over the city of Bakersfield and in the oil region, NE of Bakersfield. The enhancement over the oil region may be caused by local emissions or the wind pushing urban pollution towards the Sierra Nevada foothills. Ammonium and nitrate were typically observed in the stoichiometric ratio of ammonium nitrate. Only over the oil region, a higher ammonium-to-nitrate ratio was observed, suggesting the additional presence of sulfates. The concentration of total organics peaked in the same two regions, reaching a maximum of only 2.2 µg sm$^{-3}$. The CHARON PTR-ToF-MS instrument detected 232 organic signals (156 elementally resolved) in the 2.6 to 217.9 ng sm$^{-3}$ range, mostly associated with oxidized hydrocarbon ions ($C_xH_yO_z^+$, 59 %) and to a smaller extent with pure hydrocarbon ions ($C_xH_y^+$, 8 %). The elemental composition distribution ($C_3$–$C_{24}$, $O_0$–$O_8$; 67 % of total signal elementally resolved) is shown in Fig. S6. The complex pollution mix in the San Joaquin Valley and the relatively small organic loading make it difficult to assign selected $m/z$ signals to specific chemical compounds, and we refrain from making even tentative identifications.

During the afternoon measurements, the wind had shifted and the air temperature had increased by 2–3 °C. Figure 4b, d and f show that the boundary layer over Bakersfield and over the oil region is `CE3` now pristine. Enhanced particle pollution levels with a higher fractional abundance of organics were observed over the agricultural region SE of Bakersfield, either caused by local emissions or by the prevalent wind pushing urban pollution towards the foothills of the Tehachapi Mountains to the S of the valley. Given that we only have data from a single flight, it is not possible to carry out a more in-depth analysis of particle pollution distribution in the San Joaquin Valley. Figure 4 shall thus only be taken as an example for illustrating that airborne CHARON PTR-ToF-MS measurements can be used to map fine particulate species over regions with highly heterogeneous sources and with rapidly changing meteorological conditions.

Figure 5 shows the vertical profile data (100 m bins, shading indicates lower and upper quartiles) obtained during a missed approach at the Meadows Field Airport, which is located approximately 5 km NW of downtown Bakersfield. The data were taken between 10:48 and 10:55 local time, 17:48 and 17:55 UTC, when the planetary boundary layer extended to an altitude of 800 m a.g.l. Above the boundary layer, only organic aerosol was observed, with a maximum

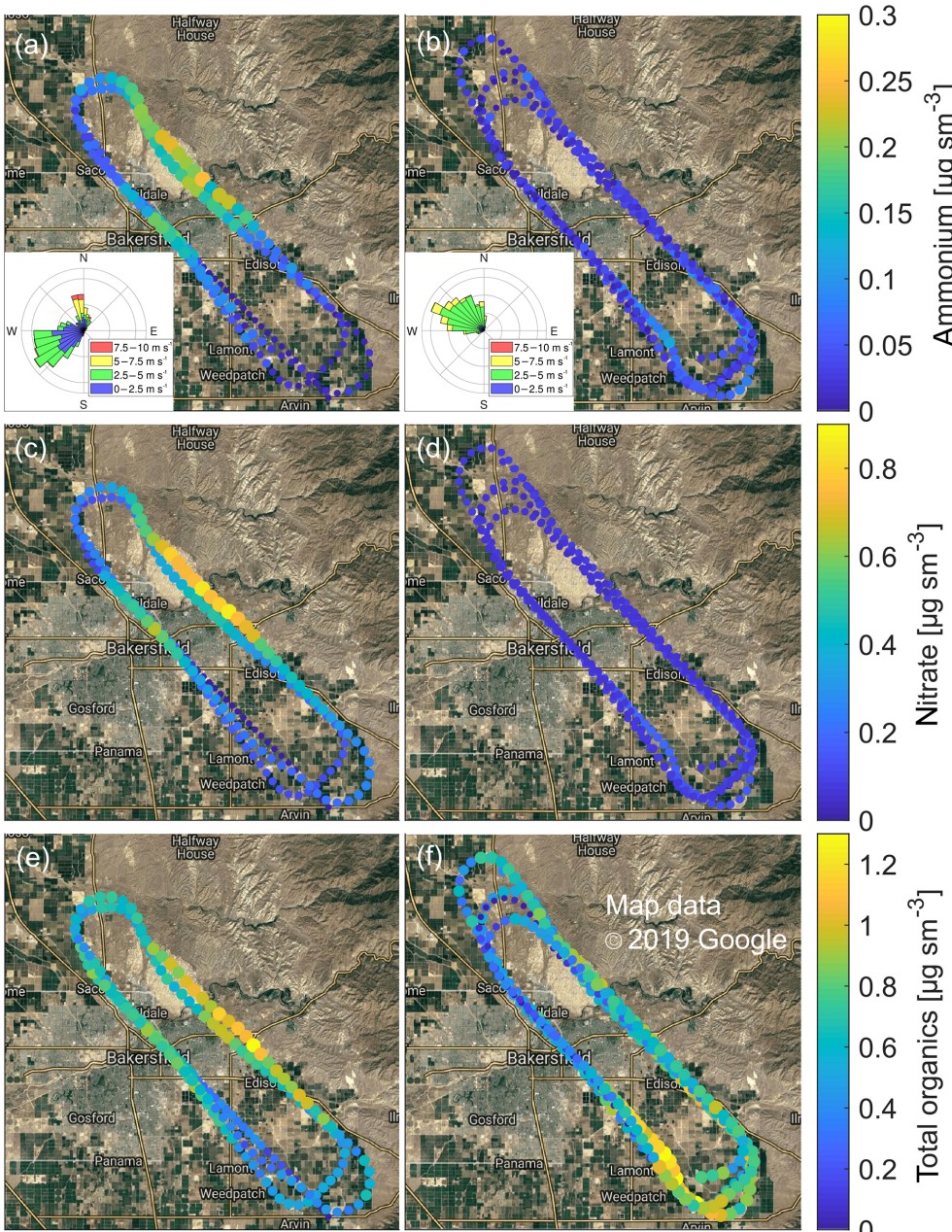

**Figure 4.** Mass concentrations of ammonium **(a, b)**, nitrate **(c, d)** and total organics **(e, f)** as observed in the boundary layer of the San Joaquin Valley in California. Individual points represent 10 s data averages. The data recorded in the late morning (10:48–11:20 local time, 17:48-18:20 UTC) are shown in **(a, c, e)**. Panels **b, d, f** refer to the early afternoon (14:02–15:01 local time, 21:02-22:01 UTC) observations. The two inserts in the uppermost panels show that the prevalent wind direction shifted from the SW to the NW between the morning and afternoon measurements.

mass concentration of $0.40 \, \mu g \, sm^{-3}$. Upon penetration into the boundary layer, ammonium, nitrate and organics exhibited a sharp increase, reaching maximum levels of 0.18, 0.19 and $0.65 \, \mu g \, sm^{-3}$, respectively, at an altitude of 500 m a.g.l.
5 The ammonium and nitrate values are in the range of mass concentrations observed by Schiferl et al. (2014) in May and June of 2010 during the CalNex field study. A close look at

the data reveals that the instrument responded faster to ammonium than to nitrate. Nitric acid is particularly sticky and thus difficult to measure at fast time response. This stickiness clearly introduced a bias in the nitrate vertical profile. 10 It is worth noting that we achieved an improved time response for nitric acid ($\tau_{1/e} \sim 10$ s) in recent laboratory experiments when a yet undisclosed treatment was applied to

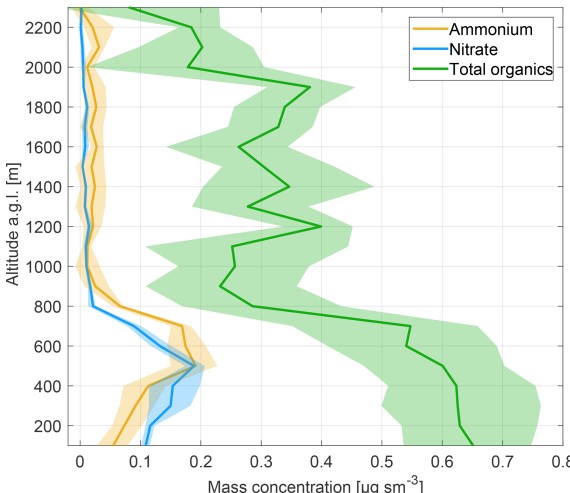

**Figure 5.** Vertical profiles (in 100 m bins, shading indicates lower and upper quartiles) of ammonium, nitrate and total organics as observed during a missed approach at Meadows Field Airport, which is located approximately 5 km NW of downtown Bakersfield, California. The data were taken in the period between 10:48 and 10:55 local time, 17:48 and 17:55 UTC.

all wetted stainless steel surfaces (Piel et al., 2019). When getting close to the ground, ammonium and nitrate concentrations dropped significantly while the concentration of total organics remained elevated. This may be explained by the fact that local sources near Meadows Field Airport are mostly urban and related to nearby oil and gas extraction and processing facilities. We again refrain from drawing any further scientific conclusions from a single missed approach to an airfield. The data are shown for exemplifying that the CHARON PTR-ToF-MS instrument can be used for vertical profile measurements.

## 4 Conclusions

We have successfully carried out the first test deployment of a CHARON PTR-ToF-MS instrument on a jet research aircraft. Most importantly, the data recorded during a test flight indicate that the instrument measures fast enough to be deployed on a jet research aircraft. The data obtained during short encounters of 3 to 15 s duration with particle plumes emanating from a small wildfire and from a refinery, respectively, demonstrate the feasibility of airborne point or small area source emission measurements. Further improvements are, however, warranted to eliminate or reduce the observed signal tailing (1/e decay time between 6 and 20 s). Recent test measurements by the instrument manufacturer with a yet undisclosed treatment of all wetted stainless steel surfaces indicate that the response time can be reduced by at least a factor of 2 compared to the data shown in this work (Piel et al., 2019). Nonetheless,

the work presented herein serves as proof of concept that CHARON can indeed be flown and generate useful data. The recorded high time resolution data allowed us to generate highly spatially resolved maps (1–2 km in the horizontal, 100 m in the vertical) of atmospheric particle chemical constituents. Exemplary data shown in this work include (i) highly time-resolved mass concentrations of ammonium, nitrate and total organics; (ii) highly time-resolved mass concentrations of classes of organic compounds (CH vs. CHO vs. CHN vs. CHNO compounds; aliphatic vs. monoaromatic vs. polyaromatic compounds); (iii) bulk aerosol average $\overline{O:C}$ and $\overline{H:C}$ ratios; (iv) highly time resolved mass concentrations of selected marker molecules (e.g., levoglucosan in particles emitted from a wildfire, an alkanolamine in particles emitted from a refinery) and (v) wildfire emission ratios ($\Delta$total organics/$\Delta$CO, $\Delta$levoglucosan/$\Delta$CO and $\Delta$vanillic acid/$\Delta$CO, $\Delta$retene/$\Delta$CO). In future deployments with a more targeted flight planning, we will be able to study changes in the particle chemical composition due to photochemical oxidation and dilution. Also, source apportionment via multivariate statistical analyses will become feasible once we obtain a comprehensive data set from an entire measurement campaign. Intercomparison measurements with more specific analytical techniques, both in the laboratory and in the field, are clearly needed to be able to better interpret the CHARON PTR-ToF-MS mass spectra and fully exploit the chemical interpretation contained therein.

The capability of the CHARON PTR-ToF-MS instrument to chemically characterize submicrometer atmospheric particles in a quantitative manner, at the near-molecular level and in real time brings a new and unprecedented measurement capability to the airborne atmospheric science community.

*Data availability.* All data are available from the corresponding author upon request.

*Supplement.* The supplement related to this article is available online at: https://doi.org/10.5194/amt-12-1-2019-supplement.

*Author contributions.* FP, MM, TM and AW designed the instrument adaptations for airborne use. FP and MM prepared the instrument for the test flight. TM and FP installed the instrument on the DC-8. FP and AW carried out the measurements. FP and MM reduced and analyzed the data. FP, MM and AW interpreted the data and conceived the paper. FP and AW wrote the paper, with comments from MM. SP provided the CO data.

*Competing interests.* Felix Piel and Markus Müller both work for Ionicon Analytik, which is commercializing CHARON PTR-ToF-MS instruments. Armin Wisthaler and Markus Müller both profit from a license agreement (CHARON inlet) between the University of Innsbruck and Ionicon Analytik.

*Acknowledgements.* Special thanks go to Barry Lefer, manager of NASA's Tropospheric Composition Program (TCP), who supported and encouraged us to test-fly our CHARON PTR-ToF-MS system. We also thank Melissa Yang Martin, Emily Schaller, Adam Webster, David Van Gilst and Steven R. Schill from the National Suborbital Research Center (NSRC) for hosting us on the NASA DC-8 during the SARP-2018 flights and for providing logistical and technical support. We further acknowledge instrumental and technical support by Bruce Anderson and Eddie Winstead from the NASA Langley Aerosol Research Group (LARGE) and by Charles (Chuck) Brock, Matt Richardson and Christina Williamson from the NOAA ESRL Chemical Sciences Division. Don Blake from UC Irvine is acknowledged for logistical support. We also thank the pilots and crew of the NASA DC-8. Our collaboration with NASA has profited from long-year financial support by the Austrian Federal Ministry for Transport, Innovation and Technology (BMVIT) through the Austrian Space Applications Programme (ASAP) of the Austrian Research Promotion Agency (FFG).

*Financial support.* FP has received funding from the European Union's Horizon 2020 research and innovation program under grant agreement no. 674911.

*Review statement.* This paper was edited by Bin Yuan and reviewed by two anonymous referees.

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

**Remarks from the language copy-editor**

CE1    "the Netherlands" is correct.

CE2    If I take the ng sm$^{-3}$ out of the brackets, I would have to write out the unit. So it would be "double-digit nanograms per standard cubic meter levels". Would you like it changed to this?

CE3    "is now pristine" is the correct tense.

**Remarks from the typesetter**

TS1    This is one of our house standards to have a space inserted within a ratio.

TS2    These changes need to be approved by the editor. Please write a statement why these changes need to be inserted and we let the editor check and confirm it. Thank you!

TS3    Please add this reference to bibliography.