# Peer review of "Airborne measurements of particulate organic matter by PTR-MS: a pilot study"

_Atmospheric Measurement Techniques, 2019_

## Referee Comment (RC1) · Anonymous Referee #1 · 25 Jun 2019

Authors present flight measurements of biomass burning and industry derived plumes observed using the CHARON+ PTRMS, each of which has been characterized on the ground and reported on by previous studies. The authors note there were some issues with the technique on its first flight deployment (notably, the high background count and the less than optimal CHARON enrichment factor), but it serves as a starting point from which further improvements can be made. There are a few areas of concern that need to be addressed before publication.

I am dubious of the aromaticity calculation given the two rather big assumptions made. Do you understand the fragmentation patterns of BB derived compounds well enough? And how do justify assumption that oxygen atoms are mostly hydroxy or carboxyl groups? The authors note that they tested fragmentation patterns on 19 acids+4 sac-

charides+2polyols, and refer details to a yet to be published article. These functional groups do not represent well the types of compounds largely found in BB plumes. How do furans fragment? How would furans alter the aromaticity calculations? How about aromatics with methoxy groups? Nitro, nitroxy or hydroperoxy groups? There is a C5 compound that is identified as a monoaromatic compound (figure S5). Is the fragmentation from the heating in the CHARON or from the PTRMS?

That is quite a bit of tailing ($\sim$6 to 20 seconds depending on I'm assuming volatility of the compound) following the interception of the BB plume (figure 1). Yes, emission ratios can still be salvaged if you sum the signal over time until signal is back to outside of plume values, if you assume these compounds interact reversibly with instrument surfaces. But this is not possible to do when you encounter numerous plumes occurring in close proximity of distinct sources. This is quite typical in biomass burning flights. What is the instrument surface that gives rise to the tailing, CHARON or PTRMS? What are your plans to mitigate this issue in the future? How much slower is the response time of nitrate compared to ammonium? More details and figures are needed characterizing the sampling performance. Suitability for flight deployment is in question.

It is surprising that organic nitrates (CHON) compose such a small fraction of the BB derived OA. Authors note that this is because plume was fresh so did not undergo much photochemistry. Please show a map as you did for the industry plume. Do you know the source of the fire? Using measured wind speed, can you estimate plume travel time or reaction time from emission to observation? Was the plume shown in figure 1 sampled when figure S2 was taken? Does not look "fresh". ON during biomass combustion can also be directly emitted, so no photochemistry needed. Please do a literature search citing range of ON fraction in OA or even total (gas + particle) carbon in fresh BB plumes (Koss et al. https://www.atmos-chem-phys.net/18/3299/2018/ comes to mind).

What is the effect of CHARON temperature? The Eichler 2015 paper (https://www.atmos-meas-tech.net/8/1353/2015/) states temperature test will be done

in future. Has that been done? Kostendiou et al (https://www.atmos-chem-phys.net/18/5799/2018/) found that BB derived OA required higher temperature than other OA types to evaporate, likely higher than 150 C. How efficiently do you suppose BB derived OA are sampled with CHARON?

What is the response time of the CO instrument? It records at 10 Hz, but the residence time in the instrument is much longer. As such, the plumes presented in figure 1 may be even sharper than what CO measurements indicate.

---

## Referee Comment (RC2) · Anonymous Referee #2 · 6 Aug 2019

General This is a report on applying the CHARON PTR-MS inlet in flight experiments. It is nicel fitting the scope of AMT. The paper shows that the time response is good enough to allow jet aircraft measurments. The paper announces the suitability of the CHARON for such measurements. There is a variety of interesting findings described. Often, it would be great when the PTR-MS information coul dbe accompanied by filter sampling and offline analysis to ad detail to the findings wwhich the PTR-MS data cannot resolve alone. Maybe that could be worth a mention.

In my vierw, the paper could be accepted subkject to only minor revision.

Details

page 6, line 27: Maybe such possible improvements could be summarized towards the

end of the paper ?

p8,l29: Is that inofmation that the three mentioned m/z are produced fm N,N-deithylethanolamin included in the references above or where is this information traceable ?

---

## Author Comment (AC1) · 5 Sep 2019

We thank Reviewer #2 for her/his timely and constructive feedback.

***Often, it would be great when the PTR-MS information could be accompanied by filter sampling and offline analysis to add detail to the findings which the PTR-MS data cannot resolve alone. Maybe that could be worth a mention.***

Unfortunately, no filter samples were collected during the SARP-2018 flights with the DC-8. A comparison of CHARON PTR-ToF-MS data with filter samples will be subject of future work.

***page 6, line 27: Maybe such possible improvements could be summarized towards the end of the paper*** *?*

We have followed this suggestion and added the following paragraph to the Conclusions: "*Further improvements are, however, warranted to eliminate or reduce the observed signal tailing (1/e-decay within 6 to 20 s). Recent test measurements by the instrument manufacturer with a yet undisclosed treatment of all wetted stainless steel surfaces indicate that the response time can be reduced by at least a factor of 2 compared to the data shown in this work (Piel et al, in preparation).*"

***page 8, line 29: Is that information that the three mentioned m/z are produced from N,N-diethylethanolamine included in the references above or where is this information traceable?***

We measured *N,N*-diethylethanolamine in our laboratory, but the data are unpublished. This is explicitly mentioned in the revised manuscript ("*unpublished data from our laboratory*").

---

## Author Comment (AC2) · 5 Sep 2019

We thank Reviewer #1 for her/his valuable comments and suggestions.

**I am dubious of the aromaticity calculation given the two rather big assumptions made. Do you understand the fragmentation patterns of BB derived compounds well enough? And how do justify assumption that oxygen atoms are mostly hydroxy or carboxyl groups?**

We have never observed the opening of an aromatic ring upon protonation in our PTR-MS instrument, which is an energetically highly unfavorable process. A substituent group on the ring (*e.g.*, alkyl, hydroxy, methoxy) may be lost upon protonation, but that does not affect the molecule's aromaticity. We thus think it is reasonable to assume that the aromaticity can be calculated even without detailed knowledge of the fragmentation patterns. The assumption that oxygen atoms are mostly bound in hydroxy or carboxyl groups is incorrect and not even needed for calculating the aromaticity equivalent. The calculation can indeed be applied to molecules that include other functional groups such as the ones mentioned by the reviewer below. We have removed this statement.

**The authors note that they tested fragmentation patterns on 19 acids+4 saccharides+2polyols, and refer details to a yet to be published article. These functional groups do not represent well the types of compounds largely found in BB plumes.**

The investigated compounds include lactic acid, 3-hydroxybutyric acid, vanillic acid, terebic acid, ketobutyric acid, maleic acid, glutaric acid, diglycolic acid, adipic acid, phtalic acid, *cis*-pinonic acid, azelaic acid, 1,3-acetonedicarboxylic acid,  $\alpha$ -ketoglutaric acid, oxaloacetic acid, malic acid, tartaric acid, citric acid, MBTCA, fructose, xylitol, levoglucosan, methyl-b-D-arabinopyranoside, erythritol, and xylose. Some of these are major constituents of BB-POA (*e.g.*, levoglucosan, vanillin, vanillic acid) but yes, the fragmentation-corrected total organic mass concentration and bulk-average *O*: *C* and *H*: *C* ratios may suffer from a bias, because the correction algorithm was not developed for this type of aerosol. This is now explicitly stated in the manuscript: "Since carboxylic and dicarboxylic acids are not the main constituents of fresh wood smoke, the data presented for the smoke plume may suffer from a small quantitative bias."

**How do furans fragment? How would furans alter the aromaticity calculations?**

Again, we have never observed the fragmentation of the furan ring nor would we expect it to fragment. What is more important with the furans is that the aromaticity equivalent calculation does no classify simple furans as aromatic. However, while furans are abundantly emitted in the gas phase (Koss et al., 2018), their relative contribution to BB-POA appears to be small (e.g., Oros and Simoneit, 2001, Schauer et al., 2001, Fine et al., 2002). Only if a substituent arm holds a pi-bonded oxygen atom (e.g., furfural or 5-hydroxymethylfurfural) and *m* is set to 0 (as in our upper limit estimate), furanoids are classified as aromatics. This information has been included in the manuscript.

**How about aromatics with methoxy groups?**

The aromaticity is not affected if the methoxy group is ejected as methanol. The detected ion is still classified as aromatic.

**Nitro, nitroxy or hydroperoxy groups?**

The aromaticity is not affected if these substituent groups are ejected upon protonation. The detected ions are still classified as aromatic.

**There is a C5 compound that is identified as a monoaromatic compound (figure S5).**

This is the  $C_5H_6NO^+$  ion, which was also observed by Laskin et al. (2009). Several aromatic structures are possible (with the N or the O atom in the ring, or the nitrosobenzenium ion fragment).

**Is the fragmentation from the heating in the CHARON or from the PTRMS?**

It is from the PTR-MS (protonation, CID in drift tube). At a vaporizer temperature of 150°C, thermal fragmentation has hitherto been observed only for a single compound (*i.e.*, an organic peroxide, R-O-O-R). This information has been added to the manuscript.

**That is quite a bit of tailing ( $\sim$ 6 to 20 seconds depending on I'm assuming volatility of the compound) following the interception of the BB plume (figure 1). Yes, emission ratios can still be salvaged if you sum the signal over time until signal is back to outside of plume values, if you assume these compounds interact reversibly with instrument surfaces. But this is not possible to do when you encounter numerous plumes occurring in close proximity of distinct sources. This is quite typical in biomass burning flights.**

We agree that the observed tailing complicates the data analysis if signals do not drop to background levels before the next plume is intercepted. We do, however, not agree that this is to be typically expected for biomass burning flights. As a matter of fact, we are currently involved in a field campaign (FIREX-AQ) in which biomass burning plumes are sampled from the NASA DC-8 in rapid sequence. Still, the time between different plume interceptions is typically 3 minutes or more which, in the case of a CHARON deployment, would be sufficient for all signals to drop to background levels. The same holds for previous airborne missions (ARCTAS, SEAC4RS) when the NASA DC-8 sampled a sequence of wildfire plumes. It may be more of a problem when operating from a small aircraft (lower airspeed, smaller turning radius) that can stay in close proximity to the plume(s). Nonetheless, this is a valid comment which we have addressed as follows: *"All signals drop to pre-plume background levels in less than 3 minutes, which is the typical time the DC-8 requires to turn and resample a plume in a meander-type flight pattern. If plumes are encountered on shorter time scales (e.g., when sampling from a small aircraft or in case fires are closely adjacent), data quality will suffer. The signal tailing needs to be eliminated or reduced to make the CHARON PTR-ToF-MS instrument more generally suitable for airborne measurements."*

**What is the instrument surface that gives rise to the tailing, CHARON or PTRMS? What are your plans to mitigate this issue in the future?**

It is the stainless steel surfaces in the vaporizer and in the drift tube. We have already described the plans for mitigating this issue in the original manuscript: "Test measurements by the instrument manufacturer with a yet undisclosed treatment of all wetted stainless steel surfaces indicate that the response time can be reduced by at least a factor of 2 compared to the data shown here (Piel et al., in preparation)" (page 6, lines 26-28). Some information is already given here: <a href="https://blog.ionicon.com/2019/03/new-ionicon-extended-volatility-range-evr-ptr-ms/">https://blog.ionicon.com/2019/03/new-ionicon-extended-volatility-range-evr-ptr-ms/</a>, but we are hesitant to include a commercial link into a scientific paper.

**How much slower is the response time of nitrate compared to ammonium?**

The nitrate signal did not increase in the observed BB plume. We suspect that this was due to the low humidity levels at 4.6 km sampling altitude, causing increased surface losses of nitric acid in the vaporizer and in the drift tube.

**More details and figures are needed characterizing the sampling performance. Suitability for flight deployment is in question.**

Unfortunately, we have obtained a limited amount of useful data during the test flight(s) and no other aerosol analyzers (*e.g.*, an AMS) were onboard the airplane allowing us to compare instrument performances and data. We thus have difficulties in providing more details and figures for characterizing the sampling performance. CHARON's performance on the ground has been characterized in previous work (Müller et al., 2017). The main two analytical challenges for deploying CHARON on an aircraft are: i) Particle sampling via an isokinetic probe coupled to pressure-stabilized inlet – here we relied on existing solutions (UH/LARGE inlet, PCI) and thus did not carry out further analytical work, and ii) an instrument response that is fast enough for capturing rapid variations in analyte concentrations. Our main goal during the test flight(s) was to test the feasibility of rapid plume sampling and urban scale surveying and four figures are given to show the potential and limitations of the CHARON PTR-MS analyzer in that respect. We agree that the analytical performance characterization given is somewhat limited but we think that even a proof-of-principle study such as our merits publication in Atmospheric Measurement Techniques.

**It is surprising that organic nitrates (CHON) compose such a small fraction of the BB derived OA. Authors note that this is because plume was fresh so did not undergo much photochemistry.**

See detailed answers below.

WORLDVIEW

Please show a map as you did for the industry plume.

The figures above shows satellite imagery (obtained on the flight day) along with the DC-8 flight track. The active fire was overflown in plume direction and the plane dropped into the plume at a safe distance, 14.3 km downwind the fire (converts into ~45 min plume travel time at the mean measured wind speed of 5 ms-1). We do not think the maps would add any significant information, but the information given in this answer has been included in the revised manuscript.

**Do you know the source of the fire?**

According to InciWeb, the Lions Fire burned in Red Fir, brush and heavy blowdown from a 2011 wind event. This information has been added to the revised manuscript.

**Using measured wind speed, can you estimate plume travel time or reaction time from emission to observation?**

See answer above.

**Was the plume shown in figure 1 sampled when figure S2 was taken? Does not look "fresh".**

The photo was taken when we approached the Sierra Nevada Mountains from the SW. It shows the older and more diluted branch of the plume to the SW (see picture above). We found a better photo showing the fresher part.

ON during biomass combustion can also be directly emitted, so no photochemistry needed. Please do a literature search citing range of ON fraction in OA or even total (gas+particle) carbon in fresh BB plumes (Koss et al. https://www.atmos-chem-phys.net/18/3299/2018/ comes to mind).

We found a mistake in the original manuscript. "The relative contributions from  $C_xH_yO_z^+$ ,  $C_xH_y^+$ ,  $C_xH_yN_z^+$  and  $C_wH_xN_yO_z^+$ ions to the total organic mass were 61%, 7%, 7% and 1%, respectively." should read "The relative contributions from  $C_xH_yO_z^+$ ,  $C_xH_y^+$ ,  $C_wH_xN_yO_z^+$  and  $C_xH_yN_z^+$  and ions to the total organic mass were 61%, 7%, 7% and 1%, respectively." The relative abundance of CHON species is thus significantly higher and comparable to what has been reported in recent comprehensive studies (*e.g.*, Jen et al. 2018).

**What is the effect of CHARON temperature? The Eichler 2015 paper (https://www.atmos-meas-tech.net/8/1353/2015/) states temperature test will be done in future. Has that been done?**

We have done some additional tests but more work is needed before these results can be published. So far, our empirical approach has been to set the vaporizer to the lowest temperature at which ammonium sulfate (AS) particles fully evaporate (*i.e.*, the ammonium signal in the PTR-MS analyzer levels off when laboratory-generated AS particles are sampled and the vaporizer temperature is gradually increased). This occurs around 150°C, which is below the temperatures at which we typically observe thermal dehydration of selected sugars. These details are now stated in the revised manuscript (Section 2.3).

**Kostendiou et al. (https://www.atmos-chemphys.net/18/5799/2018/) found that BB derived OA required higher temperature than other OA types to evaporate, likely higher than 150 C. How efficiently do you suppose BB derived OA are sampled with CHARON?**

We hopefully provided convincing evidence above that we sampled BB-POA. BB-POA is semi-volatile in nature and thus fully evaporates at 150°C (see for example Grieshop et al., 2009a,b; May et al., 2013). BB-SOA is less volatile, but also fully evaporates at 150°C if the residence time in the thermodenuder is sufficiently long to reach equilibrium (see references above). Kostendiou et al. (2018) sampled highly-aged and mixed aerosol (in which they found a BB-SOA factor) and the residence time in their denuder seems short.

**What is the response time of the CO instrument? It records at 10 Hz, but the residence time in the instrument is much longer. As such, the plumes presented in figure 1 may be even sharper than what CO measurements indicate.**

We provide a more detailed description of the CO measurement in the revised manuscript: "Carbon monoxide (CO) was measured at 10 Hz by a fast commercial  $N_2O/CO$  Analyzer (Los Gatos Research, San Jose, USA) with an observed  $2\sigma$  precision of 0.1 ppb at 1 s averaging. The flow rate was approximately 30 splm, which was achieved using a dry scroll pump (Edwards XDS35iC). The detection cell e-folding residence time was determined in the laboratory to be 0.20  $\pm$  0.05 s. The gas sampling system of the instrument was customized to maintain a constant cell pressure using a proportioning valve (IQ Valves) and PID controller (Omega) as described in Wolfe et al. (2018). Dry CO data were adjusted with a laboratory-generated calibration slope and agreed with an onboard Picarro instrument (3-s time resolution) within 3%."

**REFERENCES**

Fine, P. M., Cass, G. R. and Simoneit, B. R. T.: Chemical Characterization of Fine Particle Emissions from the Fireplace Combustion of Woods Grown in the Southern United States, Environ. Sci. Technol, 36(7), 1442–1451, doi:10.1021/es0108988, 2002.

Grieshop, A. P., Logue, J. M., Donahue, N. M. and Robinson, A. L.: Laboratory investigation of photochemical oxidation of organic aerosol from wood fires 1: measurement and simulation of organic aerosol evolution, Atmos. Chem. Phys., 9(4), 1263–1277, doi:10.5194/acp-9-1263-2009, 2009a.

Grieshop, A. P., Miracolo, M. A., Donahue, N. M. and Robinson, A. L.: Constraining the Volatility Distribution and Gas-Particle Partitioning of Combustion Aerosols Using Isothermal Dilution and Thermodenuder Measurements, Environ. Sci. Technol., 43(13), 4750–4756, doi:10.1021/es8032378, 2009b.

Jen, C. N., Hatch, L. E., Selimovic, V., Yokelson, R. J., Weber, R., Fernandez, A. E., Kreisberg, N. M., Barsanti, K. C. and Goldstein, A. H.: Speciated and total emission factors of particulate organics from burning western U.S. wildland fuels and their dependence on combustion efficiency, Atmos. Chem. Phys. Discuss., 1–22, doi:10.5194/acp-2018-840, 2018.

Koss, A. R., Sekimoto, K., Gilman, J. B., Selimovic, V., Coggon, M. M., Zarzana, K. J., Yuan, B., Lerner, B. M., Brown, S. S., Jimenez, J. L., Krechmer, J., Roberts, J. M., Warneke, C., Yokelson, R. J. and de Gouw, J.: Non-methane organic gas emissions from biomass burning: identification, quantification, and emission factors from PTR-ToF during the FIREX 2016 laboratory experiment, Atmos. Chem. Phys., 18(5), 3299–3319, doi:10.5194/acp-18-3299-2018, 2018.

Kostenidou, E., Karnezi, E., Hite Jr., J. R., Bougiatioti, A., Cerully, K., Xu, L., Ng, N. L., Nenes, A. and Pandis, S. N.: Organic aerosol in the summertime southeastern United States: components and their link to volatility distribution, oxidation state and hygroscopicity, Atmos. Chem. Phys., 18(8), 5799–5819, doi:10.5194/acp-18-5799-2018, 2018.

Laskin, A., Smith, J. S. and Laskin, J.: Molecular Characterization of Nitrogen-Containing Organic Compounds in Biomass Burning Aerosols Using High-Resolution Mass Spectrometry, Environ. Sci. Technol, 43(10), 3764–3771, doi:10.1021/es803456n, 2009.

May, A. A., Levin, E. J. T., Hennigan, C. J., Riipinen, I., Lee, T., Collett, J. L., Jimenez, J. L., Kreidenweis, S. M. and Robinson, A. L.: Gasparticle partitioning of primary organic aerosol emissions: 3. Biomass burning: BIOMASS-BURNING PARTITIONING, J. Geophys. Res. Atmos., 118(19), 11,327-11,338, doi:10.1002/jgrd.50828, 2013.

Müller, M., Eichler, P., D'Anna, B., Tan, W. and Wisthaler, A.: Direct Sampling and Analysis of Atmospheric Particulate Organic Matter by Proton-Transfer-Reaction Mass Spectrometry, Anal. Chem., 89(20), 10889–10897, doi:10.1021/acs.analchem.7b02582, 2017.

Oros, D. R. and Simoneit, B. R. T.: Identification and emission factors of molecular tracers in organic aerosols from biomass burning Part 1. Temperate climate conifers, Appl. Geochem., 32, 2001.

Schauer, J. J., Kleeman, M. J., Cass, G. R. and Simoneit, B. R. T.: Measurement of Emissions from Air Pollution Sources. 3. C 1 – C 29 Organic Compounds from Fireplace Combustion of Wood, Environ. Sci. Technol, 35(9), 1716–1728, doi:10.1021/es001331e, 2001.